# Environmental Exposure to Bisphenol A Enhances Invasiveness in Papillary Thyroid Cancer

**DOI:** 10.3390/ijms26020814

**Published:** 2025-01-19

**Authors:** Chien-Yu Huang, Ren-Hao Xie, Pin-Hsuan Li, Chong-You Chen, Bo-Hong You, Yuan-Chin Sun, Chen-Kai Chou, Yen-Hsiang Chang, Wei-Che Lin, Guan-Yu Chen

**Affiliations:** 1Department of Electrical and Computer Engineering, College of Electrical and Computer Engineering, National Yang Ming Chiao Tung University, Hsinchu 300, Taiwan; apple21038526.ee08@nycu.edu.tw (C.-Y.H.); renhao.ee09@nycu.edu.tw (R.-H.X.); eric-cyc13@nycu.edu.tw (C.-Y.C.); 2Institute of Biomedical Engineering, College of Electrical and Computer Engineering, National Yang Ming Chiao Tung University, Hsinchu 300, Taiwan; pinhsuan925.ee11@nycu.edu.tw (P.-H.L.); bohong1005.ee12@nycu.edu.tw (B.-H.Y.); cleopatra.88560@gmail.com (Y.-C.S.); 3Division of Endocrinology and Metabolism, Department of Internal Medicine, Kaohsiung Chang Gung Memorial Hospital, Chang Gung University College of Medicine, Kaohsiung 833, Taiwan; chou@cgmh.org.tw (C.-K.C.); changyh@cgmh.org.tw (Y.-H.C.); 4Department of Nuclear Medicine, Kaohsiung Chang Gung Memorial Hospital, Kaohsiung 833, Taiwan; 5Division of Neuroradiology, Department of Diagnostic Radiology, Kaohsiung Chang Gung Memorial Hospital, Kaohsiung 833, Taiwan; 6Center for Intelligent Drug Systems and Smart Bio-Devices (IDS^2^B), National Yang Ming Chiao Tung University, Hsinchu 300093, Taiwan; 7Department of Biological Science and Technology, National Yang Ming Chiao Tung University, Hsinchu 300, Taiwan

**Keywords:** Bisphenol A (BPA), endocrine-disrupting chemicals (EDCs), papillary thyroid cancer (PTC), 3D cancer spheroid, epithelial–mesenchymal transition (EMT)

## Abstract

Bisphenol A (BPA) is a prevalent environmental contaminant found in plastics and known for its endocrine-disrupting properties, posing risks to both human health and the environment. Despite its widespread presence, the impact of BPA on papillary thyroid cancer (PTC) progression, especially under realistic environmental conditions, is not well understood. This study examined the effects of BPA on PTC using a 3D thyroid papillary tumor spheroid model, which better mimicked the complex interactions within human tissues compared to traditional 2D models. Our findings demonstrated that BPA, at environmentally relevant concentrations, could induce significant changes in PTC cells, including a decrease in E-cadherin expression, an increase in vimentin expression, and reduced thyroglobulin (TG) secretion. These changes suggest that BPA exposure may promote epithelial–mesenchymal transition (EMT), enhance invasiveness, and reduce cell differentiation, potentially complicating treatment, including by increasing resistance to radioiodine therapy. This research highlights BPA’s hazardous nature as an environmental contaminant and emphasizes the need for advanced in vitro models, like 3D tumor spheroids, to better assess the risks posed by such chemicals. It provides valuable insights into the environmental implications of BPA and its role in thyroid cancer progression, enhancing our understanding of endocrine-disrupting chemicals.

## 1. Introduction

Bisphenol A (BPA) is a major component of polycarbonates and epoxy resins, which are used in the manufacture of daily life items such as pesticides, plastic compositions, food packaging, children’s toys, and medical devices [1,2]. Thus, BPA is one of the chemicals to which the human body is commonly exposed. In addition, BPA can be leached from products and transported to food, water, and air, resulting in widespread environmental contamination and human exposure [3]. BPA has been shown to leach from food and beverage containers and some dental sealants and composites under normal conditions of use [1]. Notably, over 90% of humans tested have been estimated to have measurable levels of BPA, with the highest levels being in infants and children [1,4]. Thus, BPA is also a chemical that affects the endocrine system and is hypothesized to be one of the endocrine-disrupting chemicals (EDCs) [5,6]. BPA might cause a variety of health problems including reproductive abnormalities and metabolic disorders [2,7]. Moreover, BPA has been detected in human serum, urine, amniotic fluid, follicular fluid, placental tissue, and cord blood [8]. Further, BPA might disrupt endocrine signaling pathways by mimicking hormones, affecting hormone synthesis and secretion [2,7]. As a weak ligand, BPA interferes with thyroid function by binding to thyroid hormone receptors and inhibiting their mediated transcriptional activity [4,9,10]. Moreover, the structure of BPA is similar to that of 17 beta-estradiol [11]. As a result, it regulates estrogen-responsive gene expression by competing with estrogen for binding to the estrogen receptor (ER), the androgen receptor, the membrane receptor GPER (GPR30), the epidermal growth factor receptor, and the estrogen-associated receptor [6,12].

Recent increases in thyroid cancer rates are attributed to both improved diagnostic techniques and rising environmental pollution. BPA binding to the receptor alters genomic and non-genomic signaling pathways in different cell types in varying ways, thereby influencing the biological behavior of cancer cells, particularly proliferation, invasion, growth, survival, migration, and apoptosis [12]. Research on BPA and cancer has primarily focused on female cancers, such as breast, ovarian, and cervical cancers, with evidence showing that BPA and its derivatives can stimulate ovarian cancer cell proliferation and alter cell function through changes in cellular endocrinology [13,14]. Calafat et al. [15] proposed that BPA causes structural changes in breast development and tissue and that these may result in the development of tumors. In addition, patients with a history of breast cancer, an estrogen-responsive malignancy, have been shown to have an increased risk of developing thyroid cancer [10]. Although few studies have been conducted on the effects of BPA on thyroid cancer, it has been shown to disrupt thyroid hormone homeostasis, which results in interference with normal thyroid function [4,9]. Therefore, BPA is also recognized as a thyroid disruptor [2,16]. Previous animal studies have demonstrated that exposure to environmental doses of BPA promotes thyroid cancer in immortalized rat follicular cells [17]. Ward et al. [18] compared thyroid cancer and normal thyroid cells to analyze their tolerance to BPA toxicity and found that thyroid cancer cells were more resistant than normal cells at low concentrations of BPA. Zhang et al. [19] examined how environmental chemicals regulate gene pathways in thyroid tumorigenesis using 2D cultures. The findings indicated that BPA exposure led to the HDAC6-mediated inhibition of the PTEN oncogene, which promoted tumor proliferation and migration. Zhang et al. [20] showed that BPA regulates ER to affect the proliferation and invasion of human PTC.

There are many limitations to investigating BPA in humans, including the short half-life of BPA and the possibility of simultaneous exposure to multiple chemicals, leading to confusion in the results [4]. Hence, how BPA affects the human body is still controversial. To investigate the effect of BPA on thyroid cancer cells, 2D cell models have been the dominant research method while 3D cancer models are relatively lacking. Currently, 3D cell models are widely used in drug delivery, drug sensitivity, tumor growth, and metastasis studies and have become part of a promising approach to bridging the gap between cell culture and animal models [21,22]. Three-dimensional spheroid models are increasingly used in research because they allow for the reproduction of key features of solid tumors in vivo, including intercellular signaling, extracellular matrix (ECM) deposition, and their internal material transport properties [23,24]. Cell polarity is maintained in 3D cell models; for example, in a study, specific internal structures with lumen formation could be observed in 3D adenoid spheroids [25]. Moreover, thyroid-specific proteins, including thyroid peroxidase and thyroid-stimulating hormone receptors, were found to decrease in thyroid cancer cells in the 3D culture environment. Melnik et al. [26] used microgravity to study gene and protein expression changes in follicular thyroid carcinoma (FTC) as it grows from 2D to 3D environments. The study found differences in NF-κB, hypoxia factors, epithelial–mesenchymal transition (EMT), and markers between 2D cultures and 3D spheroids. Additionally, establishing hierarchical structures of 3D tumor spheroid is of great importance for cancer research and therapy [27]. Due to the spheroids’ segmented structure and ECM barrier, the cells inside would have lower drug delivery efficacy, which is also associated with increased tumor drug resistance and resistance to toxic substances and ionizing radiation [27,28,29]. Due to the segmented structure of the spheres and the ECM barrier, the internal cells will have a lower drug delivery efficacy. This is also associated with increasing drug resistance in tumors, as well as with increased resistance to toxic substances and ionizing radiation [27,28,29]. The hypoxic and quiescent tumor microenvironment inside the spheroids may affect the expression of the sodium/iodide symporter (NIS) in thyroid cancer cells, which in turn affects their radioactive iodine (RAI) uptake and therapeutic efficacy [30]. Regarding gene expression, 3D cell models have been shown to more accurately reflect clinical gene profiles compared to 2D models, making them a promising platform for predictive clinical therapeutic testing and toxicology trials [31]. All these results illustrate the importance of using 3D models for research.

Few studies have developed 3D thyroid cancer spheroids or examined the effects of environmental toxicants like BPA, leaving its impact on PTC unclear. Therefore, this study developed a 3D papillary thyroid cancer (PTC) model to examine the effects of BPA on cell proliferation, EMT, thyroid function, and invasiveness. Additionally, since *BRAF* point mutations and *RET*/*PTC* rearrangements are common in thyroid cancer, BCPAP and TPC-1, which correspond to the above genetic mutations, were selected for this study [32]. In this study, a three-dimensional in vitro model, which was closer to the physiological environment, was adopted to investigate the effects of BPA exposure on the invasiveness behavior of thyroid cancer cells. By embedding the 3D tumor spheroids into Matrigel, which simulates the ECM, the tumor’s three-dimensional structure and cell–cell interactions were successfully reproduced, which was closer to the actual state of human tumors. The results indicated that low concentrations of BPA stimulated proliferation while high concentrations induced apoptosis in 3D PTC tumor spheroids. Following BPA exposure, thyroglobulin (TG) expression in PTC spheroids decreased with dedifferentiation and EMT. This indicated that BPA might contribute to the malignant and refractory development of PTC.

## 2. Results and Discussions

### 2.1. Establishment of 3D Thyroid Cancer Spheroids

#### 2.1.1. Analysis of Thyroid Cancer Spheroid Morphology and Growths

A 3D cancer spheroid, compared to a 2D monolayer cell culture, is characterized by the unequal supply of nutrients and oxygen leading to the formation of internal compartments, similar to the pathophysiological gradient of tumors in vivo [27,33]. In addition, 3D tumor spheroids possess cell–cell and cell–ECM interactions. This affects cellular RNA and protein expression, cell signaling, and the penetration and bioactivity of toxic substances and may further affect the overall cellular response [34].

This study selected the TPC-1, BCPAP, and MDA-T32 cell lines for experimentation as these cell lines represent subtypes derived from different mutation sites in papillary thyroid carcinoma. We hypothesize that these mutations may be associated with the malignant progression induced by BPA exposure. Specifically, the TPC-1 cell line typically harbors the *RET*/*PTC1* gene rearrangement, a mutation commonly observed in more invasive forms of papillary thyroid carcinoma [35,36]. The BCPAP cell line carries the *BRAF V600E* mutation, which is frequently linked to the early onset of papillary thyroid carcinoma [37]. Meanwhile, the MDA-T32 cell line often possesses RAS gene mutations, which are associated with tumor proliferation and development [32,38]. Therefore, three thyroid cancer cells, including TPC-1, BCPAP, and MDA-T32, were cultured in an ultra-low well plate in cell suspension to form a spheroid by a self-assembly process with a one-week culture. Imaging was performed on seven consecutive days of spheroid formation to determine tumor spheroid growth kinetics. Finally, thyroid cancer cell lines were selected for subsequent studies based on their spheroid morphology and growth status. Moreover, two cell numbers, comprising 200 and 1000 cells, were used to produce two sizes of spheroids, and the differences in cellular activity and response to toxic chemicals caused by the size difference of spheroids were compared in the subsequent experiments.

Overall, the results showed no linear increases in the volumes of both cells during spheroid formation (Figure 1a–d). The “compaction” of the cells from a loose state to a compact spheroid is a unique phenomenon observed during 3D spheroid formation, which is not usually seen in monolayer cultures [39,40]. The TPC-1 spheroid showed an irregular appearance (Figure 1a), consistent with previous studies [41,42,43]. As shown in Figure 1b, the cross-sectional area of the TPC-1 spheroid with 1000 cells decreased from the first day and reached the lowest point on the fourth day but then increased again. Similarly, the smallest cross-sectional area was observed on day 4 for a cell count of 200, with no significant changes in the cross-sectional area thereafter. As for BCPAP, its spheroid morphology tends to be round (Figure 1c), similar to the previous studies [41,42,43]. The BCPAP spheroid had its lowest cross-sectional area on the fourth day (Figure 1d). The cross-sectional area of the BCPAP spheroid with 1000 cells similarly decreased from the first day and had a minimum on the fourth day and then increased. The 200 cells also showed a minimum volume on the fourth day, which increased gradually. Therefore, we concluded that both TPC-1 and BCPAP spheroid formation occurred on the fourth day. Additionally, MDA-T32 did not show cell aggregation until the fourth day, so we suggested that it was not able to self-assemble to form a spheroid through cell suspension (Appendix A). Several reasons influence the morphology of tumor spheroids, including the cell type, culture method, and mechanical stress. In addition, tumor spheroid formation is initiated by the cytosolic integrin-mediated attachment of ECM molecules, which triggers initial aggregation [44]. Thus, we hypothesize that MDA-T32 may exhibit the reduced expression of integrins or surface adhesion molecules, leading to only loose aggregation.

#### 2.1.2. Cell Viability and Structure of Thyroid Cancer Spheroids

Live-dead cell viability assay was used to observe the changes in cellular viability within the structure over the incubation, as shown in Figure 1e. The results indicate that the whole TPC-1 spheroid was in high viability at days 4 and 7, meaning that most of it was composed of live cells. There was no viability difference between the inner and outer regions of the spheroid structure. The outer layer of the BCPAP spheroid consisted of more active regions but gradually shifted to less active dead regions in the inner layer. Moreover, the proportion of dead regions increased with spheroid size. This finding agrees with the fact that tumor spheres form a three-layered structure, consisting of an external proliferation zone, an intermediate quiescence zone, and a necrosis zone in the inner core, which explains why current studies suggest that 3D tumor spheres satisfy the requirements of cell culture models in complex in vivo environments [33,44,45].

In addition, the necrosis zone occurs gradually as the size of the spheroid increases and over time. When the spheroid diameter is less than 200 μm, it consists mainly of proliferating cells and normoxic cells, whereas a spheroid of 200–300 μm results in the three typical zones described above [27,33,44,45]. In spheroids with a diameter of about 500 μm, the formation of necrotic zones was observed [27,33]. As shown in the previous results (Figure 1a–d), the diameters of TPC-1 spheroids cultivated until day 4 ranged from about 130 μm for 200 cells to about 180 μm for 1000 cells. For the BCPAP spheroid, the range was about 160 μm for 200 cells and more than 250 μm for 1000 cells. Therefore, the high viability of the TPC-1 spheroid could be attributed to its small size, resulting in no significant oxygen and nutrient distinctions between the inside and outside of the spheroid with an overall majority of proliferating cells. Interestingly, the BCPAP spheroid developed a necrosis zone at 200 cells although its average diameter was less than 200 µm. As for the BCPAP spheroid with 1000 cells, obvious cellular zones were observed. In conclusion, TPC-1 and BCPAP exhibited significant structural differences.

### 2.2. Effect of BPA on the Growth of Thyroid Cancer Spheroid

#### 2.2.1. BPA Induces the Proliferation or Apoptosis of Thyroid Cancer

The study by Wang et al. reported that the corresponding exposure dose of BPA in the general population is <0.1 μg (kg bw)^−1^ day^−1^, which translates to approximately 10^−8^ M for an average adult weight of 70 kg [46]. Similarly, research by Li et al. found the plasma BPA exposure level in study subjects to be 10^−7^; M, with no significant differences between the healthy control group and PTC patients, indicating that this exposure level is typical of common human exposure [10]. Furthermore, Zhang et al. observed that the concentration of BPA in the urine and blood of adult women and newborns ranges from approximately 3.46 × 10^−8^ M to 5.65 × 10^−9^ M [19]. In addition, the BPA exposure encountered in daily life is typically at low doses with long-term exposure, such as via gradual accumulation through plastic containers or food packaging. However, in experimental settings, we need to use BPA doses higher than those encountered in daily life to simulate and accelerate the cumulative effects of long-term exposure, which helps in the rapid evaluation of BPA’s impact on cells or tissues. Therefore, based on the experimental parameters from the previous literature [10,19,47,48], this study employed BPA treatment concentrations ranging from 10^−4^ to 10^−9^ M, which encompass the BPA levels found in human urine or blood.

Various BPA concentrations were added to the culture medium for 24 and 48 h after thyroid tumor spheroids formed, and their effects were assessed [10,19,47,48]. The BPA treatment of thyroid cancer spheroids with varying cell counts aimed to determine whether spheroids of different sizes respond differently to BPA. Studying low-dose BPA effects also better represents environmental exposure in the human body. Figure 2a illustrates the changes in the appearance and volume of the TPC-1 spheroid composed of 200 and 1000 cells after exposure to different concentrations of BPA for 24 and 48 h. For the TPC-1 spheroid containing 200 cells, the increase in the relative cross-sectional area after exposure to various BPA concentrations followed a similar trend, with no significant differences observed (Figure 2b). In particular, the relative cross-sectional area decreased from 30.6% to 19.6% at a BPA concentration of 10^−4^ M. However, at a BPA concentration of 10^−4^ M, the relative cross-sectional area decreased from 30.6% to 19.6%. As for the TPC-1 spheroid composed of 1000 cells, the relative cross-sectional area increased after 24–48 h of exposure to different concentrations of BPA and exhibited relatively significant changes (Figure 2c). Among them, the relative cross-sectional area decreased from 43.0% to 33.4% when the BPA concentration was 10^−4^ M. However, at a BPA concentration of 10^−5^ M, the relative cross-sectional area increase rate increased from 36.1% to 40.0%. In addition, the TPC-1 spheroid with 1000 cells had a more considerable relative cross-sectional area increase than that with 200 cells.

The appearance of the BCPAP spheroid after exposure to BPA for 24 and 48 h is shown in Figure 3a. The BCPAP spheroid remained roughly circular with no significant change. Interestingly, the 10^−4^ M BPA caused the BCPAP spheroid to become loose on the exterior, becoming more pronounced over time. The cross-sectional area of the spheroid with 200 cells increased after 24 and 48 h of exposure to 10^−9^–10^−4^ M BPA (Figure 3b). Nonetheless, the spheroid composed of 200 cells showed an increase in volume of about 137.9% after 24 h of exposure to 10^−4^ M BPA compared to that of the unaddressed group, but only about 114.9% after 48 h. The spheroid consisting of 1000 cells showed an upward trend in the cross-sectional area after exposure to different concentrations of BPA (Figure 3c). In addition, a greater concentration of BPA resulted in a greater volume change rate for the BCPAP spheroid. Notably, the spheroid with 1000 cells displayed an apparent loose structure in the periphery after adding 10^−4^ M BPA. General agreement exists on the possible association between BPA and thyroid nodules or thyroid cancer [4]. It is known that BPA triggers phosphoinositide 3-kinase (PI3K) and mitogen-activated protein kinase (MAPK) by binding to the ER in cells. This activates the cell proliferation pathway’s downstream pathways, thus affecting cell growth [2,4,16,20]. Moreover, the concentration of BPA causes cell proliferation, growth inhibition, and even cell death [10,19,20].

To further understand the reasons for the volume changes and structural changes of the two spheroids, we next examined the cells by live/dead cell viability assay. No low viability core was present in the TPC-1 spheroid with both cell counts (Figure 4a). When TPC-1 spheroid was exposed to 10^−6^ M and 10^−5^ M BPA, it mainly showed calcium-AM, labeled in green. This result indicated that these two concentrations of BPA did not have significant effects on the cellular viability of the TPC-1 spheroid. However, upon exposure to 10^−4^ M BPA, TPC-1 spheroids with 200 cells maintained high cellular activity while those with 1000 cells began to exhibit cell death. In the absence of BPA, the structure of the BCPAP spheroid could be divided into a proliferative zone with higher viability in the outer hemisphere and a necrotic zone with lower viability in the inner nucleus (Figure 4b). The spheroid structure at lower concentrations of BPA (10^−6^ M and 10^−5^ M) did not differ much from that without BPA. We propose that the two lower BPA concentrations may stimulate cell proliferation in the outer layer of the spheroid, increasing its volume, while the core necrotic region remains unaffected. Nevertheless, when the BCPAP spheroid was treated with a higher concentration (10^−4^ M) of BPA for 24 h, the boundary between the viable and dead regions of the cells disappeared. After 48 h of treatment, the overall spheroid was almost entirely labeled by PI, which means that only a few cells were alive, representing a significant decrease in the viability of the BCPAP spheroid. We thus speculated that the decline in the BCPAP spheroid cross-sectional area due to 10^−4^ M BPA treatment was probably caused by the decrease in peripheral cell viability. 

After the addition of 10^−4^ M BPA, the BCPAP spheroid showed a loose peripheral structure (Figure 3a), and the cell viability of these outer layers was not high (Figure 4b). This was likely attributed to the fact that BPA induces apoptosis and necrosis in cancer cells [48,49]. Therefore, the increase in the BCPAP spheroid volume at higher concentrations (10^−4^ M) of BPA resulted from the reduced cell viability of its outer layers, leading to an unstable spheroid structure. Additionally, when the BPA dose was raised to 10^−4^ M, the external layer of the BCPAP spheroid with cell number 1000 started to show a loosened state (Figure 3a), which was more evident after 48 h, suggesting that the BPA might affect the cells from the external layer of the spheroid. From Figure 1e,f, it is obvious that the cellular active structures of TPC-1 and BCPAP spheroids differed significantly, with the latter having a central necrotic zone, which could also affect the penetration of chemical substances. Zhang et al. [19] showed that 5 × 10^−5^ M BPA already inhibited the cellular viability of 2D BCPAP. This might be because the cellular structure of 3D cells differs in the penetration of chemicals, including in the penetration rate and penetration depth. Thus, the 3D spheroid showed better tolerance to chemicals, demonstrating the importance of utilizing 3D cellular models for drug studies and dosage estimation. 

When the spheroid was exposed to relatively low concentrations of BPA (10^−9^–10^−5^ M), it showed an elevated volume but no significant change in cellular activity. Similarly, Li et al. [10] reported that 2D thyroid follicular epithelial cells (Nthy-ori 3-1) treated with 10^−6^ M and 10^−7^ M BPA showed remarkable proliferation whereas 10^−4^ and 10^−3^ BPA inhibited cell growth. Zhang et al. [20] proposed that low concentrations (10^−6^–10^−8^ M) of BPA could promote the proliferation of PTC (BHP10-3) by modulating ERs. Therefore, this concentration of BPA could induce the proliferation of BCPAP and TPC-1 spheroids, with the former being the more pronounced. In addition, the TPC-1 spheroid exhibited cell death after 24 h, and the number of apoptotic cells increased after 48 h of exposure to a higher concentration (10^−4^ M) of BPA. While the BCPAP spheroid showed a significant decrease in cellular activity, we concluded that the TPC-1 spheroid might have a higher tolerance to BPA than BCPAP.

#### 2.2.2. Morphological Analysis of F-Actin

The cytoskeletal differences between thyroid cancer cells may lead to the different abilities of the invasion and formation of 3D structures [50]. The outer layer of the BCPAP spheroid showed a visibly loose appearance when exposed to BPA (Figure 3a). Therefore, BPA might be responsible for the disruption of the BCPAP spheroid structures such as the tumor spheroid skeleton or cell–cell junctions (Figure 5. Non-BPA-treated BCPAP spheroids exhibited well-arranged cell margins and a reticulated cellular pattern. The F-actin was well expressed and the cells were tightly connected. On the other hand, in BPA-treated BCPAP spheroids, not all the cells showed a complete cytoskeleton, being primarily expressed in the surrounding area of a single cell. Most importantly, cell-to-cell connectivity was not observed, and F-actin was disrupted.

Similarly, BPA also diminished F-actin expression in the proximal tubule of renal tissue, further leading to developmental abnormalities and compromising physiological function [51]. Previous studies have shown that BPA altered cell morphology, function, and motility [48,52]. Actin plays a crucial role in various cellular processes, such as cell motility, division, and signaling, and is essential for maintaining cell morphology and establishing cell-to-cell connections. [52]. The reorganization of the cytoskeleton is an essential step for tumor cell motility. After the disintegration of the actin cytoskeleton in abnormal cells, the ratio of the nucleus to the cytoplasm is subsequently altered, ultimately contributing to tumor formation and metastasis [53,54]. Moreover, an increase in pericellular F-actin expression indicates altered actin dynamics, which may drive malignant cells to a more aggressive phenotype [54,55,56]. The above results implicated that the malignant progression of cancer caused by BPA by affecting F-actin expression should not be overlooked.

### 2.3. Effects of BPA on Thyroid Function in PTC

#### 2.3.1. TG Expression in 2D Thyroid Cancer Cells

TG participates in thyroid hormone synthesis as a large glycoprotein bound to iodine and produced by normal thyroid follicle cells and differentiated thyroid carcinomas [4,57]. Therefore, TG serves as a sensitive marker for thyroid function in diagnostic pathology [58,59,60,61]. Since TG is produced and utilized exclusively in the thyroid gland, it acts as an ideal tumor marker clinically [59,60]. Moreover, elevated TG expression is one of the indicators of thyroid cancer; measuring TG can be used to monitor the prognosis of thyroid cancer patients and their recurrence [59,60,62,63]. Under the condition of 2D culture, the cytoplasm of both BCPAP and TPC-1 expressed TG (Figure 6a). These indicated that these cell lines used in this study possess a certain degree of differentiation characteristics and thyroid functions.

#### 2.3.2. BPA Resulted in Decreased TG Expression of PTC Spheroids

Next, we analyzed the TG expression of 3D PTC spheroids. The results showed that both BCPAP and TPC-1 spheroids expressed TG and displayed a homogeneous expression of the inner and outer spheroid layers, similar to previous studies [45] (Figure 6b,d). However, following the addition of BPA, TG expression in both cases decreased significantly, with the fluorescence intensity per unit area being reduced by approximately 1.69-fold and 3-fold, respectively (Figure 6c,e). TG expression is associated with the extent of cellular differentiation, making it an important marker for thyroid function and cellular differentiation [57,59,60,61]. Thyroid cancers are classified into differentiated thyroid cancer (DTC), progressively dedifferentiated DTC, poorly differentiated thyroid cancer (PDTC), and anaplastic thyroid cancer (ATC) according to their histopathological differentiation [59,60,61]. PTC is one of the differentiated cancers [32,64,65]. Moreover, the level of TG expression in thyroid cells correlates with their degree of differentiation [57,59,60,61]. PTC cells usually express significantly lower amounts of TG than normal thyroid tissue [57,61]. In terms of prognosis for patients with thyroid cancer, DTC has the best prognosis, but APC can be difficult to cure and is often associated with fatality [65]. Furthermore, dedifferentiation and EMT are associated with cancer progression and metastasis as they induce cancer cells to change from a poorly differentiated to a dedifferentiated state [61,66]. As PTC progresses to ATC through dedifferentiation, it also explains the poor prognosis of ATC patients [66]. The dedifferentiation of DTC leads to a loss or reduction in TG expression and iodine uptake, thereby diminishing the efficacy of or potentially causing resistance to RAI therapy [65,67]. Consequently, although both BCPAP and TPC-1 are PTC cell lines that were studied, BCPAP showed a more pronounced decrease in TG, possibly due to its origin from poorly differentiated PTC [32,68]. Though there is currently limited research on whether BPA exposure affects the efficacy of standard treatments, such as iodine-131 (I-131) therapy, studies indicate that BPA may interfere with thyroid hormone signaling [4,69,70], induce EMT [10,71], and impact thyroid differentiation markers such as the NIS [72,73]. Moreover, BPA may inhibit iodine uptake by interacting with the NIS [72]. I-131 therapy relies on the NIS to transport radioactive iodine into thyroid cells for the destruction of hyperactive thyroid tissue or thyroid cancer cells [74,75]. Therefore, we hypothesize that if BPA alters NIS function, it could potentially reduce iodine-131 uptake efficiency, leading to therapeutic resistance in thyroid cancer cells and affecting treatment precision. In conclusion, BPA may affect dedifferentiated papillary thyroid carcinoma, further affecting its TG expression and even lowering its iodine uptake, resulting in the therapeutic efficacy of resistance to RAI, while future studies are needed to explore the interaction between BPA and I-131 therapy to assess these potential effects.

### 2.4. Effect of BPA on Cancer Characteristics of PTC

#### BPA Promotes the Occurrence of EMT in PTC

Cancer metastasis is one of the causes of mortality in malignant cancers while the EMT of cancer cells is considered an essential step in cancer migration and invasion [76,77]. EMT is typical in invasive thyroid cancer and associated with the progression of thyroid cancers including well-differentiated, poorly differentiated, and anaplastic thyroid carcinomas [78,79,80]. Thyroid cancer cells exhibit morphological and molecular changes during this process. Epithelial cells lose their polarity and intercellular adhesion. Therefore, epithelial features such as E-cadherin and EpCAM are downregulated whereas mesenchymal features such as vimentin and matrix metalloproteinase 9 (MMP 9) are upregulated [76,78,81,82]. Such processes ultimately contribute to the reduction of cell–cell adhesion and the acquisition of enhanced motility [83,84]. Thus, EMT activation plays an integral role in thyroid cancer by promoting capsular invasion and remote metastasis [85]. A decrease in or loss of E-cadherin expression and increased vimentin expression are hallmarks of EMT [79,83]. In thyroid cancer progression, E-cadherin and vimentin can be considered biomarkers related to aggressive, poorly differentiated, and malignant phenotypes [26,85]. Therefore, we first examined the expression of the proteins TPC-1 and BCPAP. Following this, the impact of BPA on the EMT of TPC-1 and BCPAP was analyzed using E-cadherin and vimentin as markers.

The E-cadherin expression of TPC-1 and BCPAP was not prominent under 2D monolayer culture conditions (Figure 7a). However, under 3D spheroid culture conditions, TPC-1 spheroids exhibited increased levels of E-cadherin mainly distributed in the periphery of the spheroids (Figure 7b). After exposure to BPA, the E-cadherin expression of the TPC-1 spheroid decreased significantly (Figure 7c). As for the BCPAP spheroid, the E-cadherin expression was still not significant (Figure 7d). With the addition of BPA, the E-cadherin expression of the BCPAP spheroid reduced (Figure 7e). Vimentin, as an intermediate filament, has been linked with accelerated cell growth, invasion, and poor prognosis in oncology studies [85,86]. Both TPC-1 and BCPAP cells exhibited vimentin under 2D monolayer culture conditions (Figure 8a). This might suggest that they have potential for EMT and are metastatic thyroid cancer cells. TPC-1 spheroid showed vimentin expression before BPA exposure, which increased after BPA exposure (Figure 8b,c). Similarly, the BCPAP spheroid also showed higher vimentin expression after BPA treatment (Figure 8d,e). 

Overall, in 2D culture, the expression of E-cadherin in TPC-1 and BCPAP spheroids was relatively low whereas the expression of vimentin was high. However, in 3D culture, there was a significant increase in the expression of E-cadherin in the TPC-1 spheroids and vimentin in the BCPAP spheroids. The expression of E-cadherin and vimentin is affected by the 2D or 3D culture environment, indicating that the adhesion strength and polarity of adherent junctions between PTC cells could be altered by the cellular structure [25]. Furthermore, BPA-induced EMT in PTC spheroids caused a marked decrease in E-cadherin and an increase in vimentin, with E-cadherin reduction being more pronounced in TPC-1 spheroids and vimentin increase being more evident in BCPAP spheroids. Since both spheroids showed the abovementioned protein expression changes after 24 h of 10^−4^ M BPA exposure, this experimental concentration might induce a higher invasiveness of PTC cells.

Epithelial cells connect through tight junctions and adherens junctions, with E-cadherin belonging to the latter [87]. Given that E-cadherin plays a critical role in intercellular adhesion, it is crucial for inhibiting cell proliferation and preventing invasion. Consequently, E-cadherin is not only considered a tumor suppressor protein but is also linked to stemness in three-dimensional spheroid structures [45,87]. In thyroid cancer, the presence of E-cadherin is correlated with a better prognosis [26,85]. Additionally, BCPAP spheroids showed a deficient expression of E-cadherin compared to TPC-1 (Figure 8b). This might suggest that BCPAP spheroids do not have strong intercellular connections, possibly explaining why BPA penetrates more easily or rapidly into BCPAP spheroids than into TPC-1, causing the whole tumor spheroid to enter a less active state (Figure 4). This also explains why the outer cellular structures of BCPAP spheroids became looser after the addition of BPA (Figure 3a). Vimentin expression levels vary in stages as cancer progresses. Initially, vimentin concentrations are deficient, but they increase as cancer cells invade the surrounding tissue [88]. In addition, vimentin upregulation renders the spheroid structure less dense and unstable, leading to higher invasiveness and motility in subsequent cells [45]. For example, high vimentin expression has been associated with PTC invasion and even lymph node metastasis [79]. Also, BPA exposure has been shown to cause disconnected F-actin and increased EMT-associated protein expression in lung cancer cells [52]. Additionally, vimentin is required for tumor progression and metastasis in lung cancer [86]. When cells undergo EMT, their intracellular skeleton is remodeled, and intercellular connections are weakened or lost [83]. Combining Figure 4 with the above results, BPA administration could induce PTC cells to detach from the epithelial cell population more efficiently and further migrate and invade the surrounding tissues. This is of great significance in the cancer metastasis and invasion process.

In addition to E-cadherin and vimentin, proteins such as N-cadherin [10,52], fibronectin [52], Snail/Slug/Twist [71], MMPs [10,89], and TGF-β [90,91,92] also play key roles in the EMT process of thyroid cancer. BPA may promote thyroid cancer progression and metastasis by modulating these proteins.

### 2.5. Effects of BPA on Thyroid Cancer Spheroid in ECM Model

#### 2.5.1. Invasion Assay of 3D Thyroid Cancer Spheroids into Matrigel

ECM is a complex extracellular structure composed primarily of collagen I, designed to provide cellular support and facilitate cellular functions including adhesion and migration [45,93]. In tumor metastasis, cancer cells detach from the primary tumor, reorganize their cytoskeleton and adhesion properties, and degrade the surrounding matrix to invade adjacent tissues [94]. Thus, investigating the invasive behavior of cancer cells with 3D models provides more information on cell–matrix interactions and matrix remodeling [93]. When simulating ECM with Matrigel, the appropriate concentration varies depending on the method of use and the cell line. The concentration of hydrogel indirectly affects matrix stiffness, which then influences cell behavior. To study the invasiveness of thyroid tumor spheroids while limiting the effects of growth factors in Matrigel, we used a growth-factor-reduced Matrigel to simulate the ECM. We embedded a single thyroid tumor spheroid in a 3D Matrigel to observe its invasion [95,96]. To assess the impact of matrix concentration on cell invasion, we used Matrigel at 3 mg/mL and 5 mg/mL and monitored cell invasion over seven days (Figure 9 and Figure 10). The maximum cross-sectional area was measured with an inverted microscope, and the percentage of cell invasion was calculated by comparing the invasion area on each day to the initial area on day 0.

TPC-1 spheroid embedded in two concentrations of Matrigel showed radiation of invasion (Figure 9a,b). Spheroids with 200 and 1000 cells showed a gradual increase in invasion area over time, with the 1000-cell spheroids exhibiting more pronounced invasion (Figure 6c,d). The invasion behavior of the BCPAP spheroid in Matrigel was also oriented in all directions (Figure 10a,b). Likewise, the invasion area of the BCPAP spheroid increased with the number of days. BCPAP spheroids with cell number 200 exhibited significant invasion after being embedded in Matrigel on the third day in both concentrations (Figure 10c). For the cell number 1000 of the BCPAP spheroid, the invasion area increased dramatically from the second day (Figure 10d). Its invasion area increased markedly compared to that of the smaller ones. These results indicated that both spheroids exhibited a significant increase in the percentage of invasion area at a Matrigel concentration of 3 mg/mL. We proposed that higher Matrigel concentrations increased stiffness and adhesion, leading to reduced ECM degradation, slower migration, and a smaller invasion area [28,84]. In addition, the percentage increase of invasion area of TPC-1 was higher than that of BCPAP, indicating that TPC-1 had a higher degradation capacity of ECM or a faster migration rate. 

Interestingly, the TPC-1 spheroid exhibited a “starburst” invasion pattern in Matrigel whereas the BCPAP spheroid exhibited a “budding” (Appendix A). The invasion pattern of tumor spheroids in the 3D invasion assay was not uniform; for example, the human highly malignant glioblastoma cell line (U-87 MG) and human triple-negative breast cancer cells (MDA-MB-231) presented similarly to the TPC-1 spheroid. In contrast, human squamous head and neck cancer and breast cancer cells (MCF 7) appeared to resemble the BCPAP spheroid [97,98].

#### 2.5.2. Effect of BPA on the Invasion of PTC Spheroids in the ECM Model

Finally, two thyroid spheroids were embedded in 3 mg/mL Matrigel to investigate the induction of BPA on their invasion ability. The cell invasion percentage was calculated as the increase in invasion area relative to day 0, accounting for BPA exposure in the Matrigel-embedded spheroid. The TPC-1 spheroid exhibited different levels of invasion over time induced by various concentrations of BPA (Figure 11a,b). When 200 cells were present, 10^−4^ M BPA resulted in the maximum extent of invasion (Figure 11c,d). From the third day onwards, the 10^−4^ M BPA group showed a significantly higher increase in the invasion area than the other lower-concentration ones. There were no considerable differences between the results at the lower concentrations of 10^−6^ M and 10^−5^ M and the control group. Although the TPC-1 spheroid showed a decrease in cellular viability when exposed to 10^−4^ M BPA, some of its cells remained active (Figure 4a). Accordingly, BPA might induce TPC-1 to develop the ability to degrade ECM and to show a higher invasion.

The BCPAP spheroid showed varying degrees of invasion over time induced by different concentrations of BPA (Figure 12a,b). The BCPAP spheroid of 200 cells exposed to higher concentrations of BPA (10^−5^ M and 10^−4^ M) experienced an increase in the extent of invasion from day 1 to 5 but then decreased (Figure 12c). The 10^−6^ M BPA treatment gradually increased the invasion extent of the BCPAP spheroid over time. Only the highest concentration of 10^−4^ M BPA in the BCPAP spheroid containing 1000 cells showed a trend of initially increasing and then decreasing invasion (Figure 12d). Both 10^−6^ and 10^−5^ M BPA treatments caused a gradual increase in the invasion of the BCPAP spheroid over time. As shown in Figure 4, the BCPAP spheroid exposed to 10^−4^ M BPA for 48 h displayed a noticeable decrease in cellular viability. Therefore, we considered that the sudden reduction in the invasion range could be due to the apoptosis of the spheroid on the fifth day or so, which reduced the invasive ability. Moreover, a spheroid composed of 1000 cells might be larger in size and, therefore, more tolerant to BPA.

Overall, the TPC-1 spheroid was more tolerant to BPA than the BCPAP spheroid (Figure 4). This could explain why a higher concentration (10^−4^ M) of BPA could induce a more pronounced invasion range of the TPC-1 spheroid. The BCPAP spheroid was more sensitive to BPA, increasing its invasive ability at lower concentrations. Similarly, human ovarian adenocarcinomas (SKOV3 and A2780) showed enhanced EMT and invasion when exposed to relatively low concentrations of BPA (10 or 100 nM) [76]. Low quantities of BPA are likely to promote protein degradation in the ECM by upregulating MMP9. This also occurs in triple-negative breast cancer [76]. In addition, Figure 5 shows that both TPC-1 and BCPAP spheroids could generate EMT induced by BPA. Compared with epithelial cells, mesenchymal cells showed stronger vimentin. When cells turn to express more vimentin, this indicates a shift towards a mesenchymal phenotype, possibly with higher motility and aggressiveness [88,99]. Consequently, the results in Figure 7 correspond to the increased invasiveness of the two TPC-1 and BCPAP spheroids when exposed to BPA.

On the other hand, the differential responses of TPC-1 and BCPAP cell lines to BPA might be influenced by their genetic and epigenetic factors. Regarding genetic mutations, the genes most commonly affected in PTC are *BRAF* and *RET*, which are activated through point mutations or chromosomal rearrangements [100]. The TPC-1 cell line typically harbors the *RET*/*PTC1* gene rearrangement, a mutation frequently observed in more aggressive forms of PTC [35,36,101]. The *RET* gene mutation leads to the sustained activation of downstream signaling pathways, particularly the MEK/ERK and PI3K/AKT pathways, often resulting in abnormal cell proliferation, survival, and anti-apoptotic effects [102]. In contrast, the BCPAP cell line contains the *BRAF V600E* mutation. This mutation causes the persistent activation of the MAPK/ERK pathway, which promotes cell proliferation and growth, and is commonly associated with the early onset of PTC [37]. Moreover, this mutation induces the upregulation of tissue inhibitor of metalloproteinases-1 (TIMP-1), thereby facilitating PTC proliferation and metastasis [35]. However, unlike the *RET* mutation, the *BRAF V600E* mutation primarily affects the MEK/ERK pathway [103,104] without directly impacting the PI3K/AKT pathway [105,106,107].

Furthermore, epigenetic abnormalities are present in almost all cancers and, in conjunction with genetic alterations, promote tumorigenesis [108]. In thyroid cancer cells, epigenetic modifications encompass DNA methylation, histone modifications, and the expression of non-coding RNAs, with therapeutic agents targeting the first two modifications currently under clinical investigation [107,109]. *BRAF* mutations, in particular, are closely linked to the abnormal methylation of multiple tumor suppressor genes including TIMP3. Research suggests that the methylation-induced silencing of TIMP3 plays a critical role in the invasion and progression of PTC driven by *BRAF* mutations [108]. Additionally, PTEN, a phosphatase that inactivates the PI3K/Akt pathway, exhibits abnormal methylation in approximately 50% of PTC and almost all follicular carcinomas and adenomas, implicating its potential involvement in thyroid tumor development [110]. Given the potential epigenetic differences between TPC-1 and BCPAP cell lines, these variations may influence the cellular response to BPA, resulting in distinct gene expression profiles that alter the effects of BPA.

We will subsequently discuss the induction of EMT by BPA and its potential to enhance cellular invasiveness with a particular focus on the underlying molecular mechanisms involved. BPA influences the EMT process through multiple signaling pathways, which interact and synergistically induce EMT, thereby increasing cell invasiveness and metastatic potential. (1) BPA activates the PI3K/Akt signaling pathway, which plays a critical role in cell proliferation, survival, and migration. The overactivation of this pathway leads to an increase in the number of MMP proteins, promoting matrix degradation and enhancing cell invasiveness [12,20,111]. (2) BPA can activate the MAPK/ERK pathway, which induces changes in the cytoskeleton and the expression of MMPs such as MMP-2 and MMP-9, further strengthening cell migration and invasiveness [12,111,112]. (3) BPA enhances the activity of TGF-β, a key EMT inducer, promoting the expression of EMT-related proteins like N-cadherin and vimentin, which facilitates the transition of cells from an epithelial to a mesenchymal phenotype [113,114].

Specifically, research indicates that the MAPK and PI3K/Akt pathways are critical for cancer cell proliferation, survival, and migration [111,115,116]. BPA can activate the PI3K/Akt pathway, enhancing these processes [12,111], and the dysregulation of this pathway is common in thyroid cancer [12,20]. BPA also modulates the MAPK pathway, promoting EMT and invasiveness [10,12,112]. For example, the BPA activation of the MAPK/ERK and PI3K/Akt pathways increases MMP-2, MMP-9, and N-cadherin expression, enhancing ovarian cancer cell migration [111]. Studying these pathways under BPA exposure could deepen our understanding of cancer biology and identify potential therapeutic targets or biomarkers.

Last, exposure to BPA within the tumor microenvironment may alter the interactions between tumor cells and stromal cells such as fibroblasts and immune cells, thereby affecting tumor progression [117], promoting an immunosuppressive environment [118,119], and enhancing immune cell recruitment [120,121,122]. Specifically, BPA has been shown to facilitate the polarization of macrophages toward the M2 phenotype [118], reduce the recruitment of regulatory T cells [120], and activate fibroblasts into tumor-associated fibroblasts [118,123]. Thus, analyzing BPA’s effects on tumor cell interactions with tumor-associated fibroblasts or immune cells in a 3D microenvironment is vital for understanding its impact on tumor progression.

## 3. Materials and Methods

### 3.1. Cell Culture

This study included human papillary thyroid carcinoma cell lines: MDA-T32, BCPAP, and TPC-1. BCPAP (DSMZ) and TPC-1 (obtained from Jacques Dumont at Université Libre de Bruxelles) were cultured in RPMI-1640 (ATCC, #30-2001, Manassas, VA, USA), as was MDA-T32 (ATCC, #CRL-3351, Manassas, VA, USA). The medium was supplemented with 10% fetal bovine serum (Corning, 35-010-CV, Corning, NY, USA) and 1% penicillin–streptomycin (HyClone, #SV30010, Logan, UT, USA). Cells were incubated at 37 °C in a 5% CO_2_ incubator. When the cells reached 80–90% confluence, they were passaged using 0.25% trypsin protease (HyClone, SH30042.01, Logan, UT, USA) for cell resuspension and subculturing.

### 3.2. Spheroid Cultures

Spheroids were generated through the self-assembly process of cell suspensions. After counting the cell number, they were suspended in the appropriate culture medium and seeded into a 96-well clear round-bottom ultra-low-attachment microplate (Corning, #7007, Corning, NY, USA). Each well formed a single spheroid. The plate was then incubated at 37 °C with 5% CO_2_ to promote spheroid formation. Images of the spheroids were captured and recorded every 24 h using an inverted microscope at random fields of view. The cross-sectional area of the spheroids was quantified using ImageJ (1.54 h) and the growth curves were plotted.

### 3.3. BPA Exposure

BPA (Sigma-Aldrich, #239658, Burlington, MA, USA) solutions were prepared according to the manufacturer’s instructions. The BPA solution was diluted and added to BCPAP and TPC-1 spheroids on the fourth day, marking this as day 0 of BPA treatment. The spheroids were then imaged every 24 h using an inverted microscope at random fields of view to capture and record the maximum cross-sectional area. The effects of BPA concentrations at 10^−9^ M, 10^−6^ M, 10^−5^ M, and 10^−4^ M on the cells were evaluated at exposure times of 24 h and 48 h. Finally, ImageJ was used to quantify the cross-sectional area.(1)$$cell\;relative\;area\;\left(\%\right)=\frac{Area\;\left(Day\;X\right)-Area\;\left(D0\right)}{Area\;\left(D0\right)}\times 100\%$$

### 3.4. Spheroid Embedding in the ECM Model

To simulate the interaction between the ECM and cells, spheroids were embedded in Matrigel to assess their invasive capabilities. Matrigel (Biocoat, #356231, Horsham, PA, USA) was diluted in ice-cold Dulbecco’s Phosphate-Buffered Saline (DPBS) to a concentration of 3 or 5 mg/mL and gently added to the wells of a 96-well clear round-bottom ultra-low-attachment microplate (Corning, #7007, Corning, NY, USA). After incubation at 37 °C for 2 h, culture medium (with or without BPA) was added on top. Observations and bright-field image documentation were performed using an inverted microscope. Images of cells invading from the spheroid were captured starting from day 1, with intervals extending up to 7 days. The extent of invasion was analyzed by measuring the maximum cross-sectional area of each spheroid.(2)$$cell\;invasion\;\left(\%\right)=\frac{Area\;\left(Day\;X\right)-Area\;\left(D0\right)}{Area\;\left(D0\right)}$$

### 3.5. Live/Dead Cell Viability Assay

A live/dead cell assay kit (ABP Biosciences, #A017, Rockville, MD, USA) was used to analyze the viability of cells within the tumor spheroids, following the manufacturer’s instructions. Briefly, propidium iodide (PI), which stains dead cells, and Calcein-AM (AM), which stains live cells, were diluted in DPBS and mixed thoroughly by vertexing. The diluted dyes were then added to the samples for analysis and incubated in a cell culture incubator protected from light. After incubation, the dye solution was removed, and the samples were washed several times with DPBS. Fluorescent images were captured using a fluorescence microscope with FITC (Calcein-AM) and TRITC (PI) channels and the images were merged.

### 3.6. Cytoskeleton Analysis

After removing the culture medium, the cells were fixed with 4% formaldehyde and then stained for the filamentous actin (F-actin) using Alexa Fluor™ 488 phalloidin (ThermoFisher, #A12379, Waltham, MA, USA). Images were captured using a confocal laser scanning microscope (LEICA, TCS SP8, Wetzlar, Germany).

### 3.7. Immunofluorescence Staining

After removing the culture medium, the cells were fixed with 4% formaldehyde and permeabilized with 0.01% Triton X-100. All steps were followed by washing with DPBS. Primary antibodies TTF-1 (Cell Signaling, #12373, Danvers, MA, USA), TG (Abcam, AB156008, Cambridge, UK), E-cadherin (Abcam, AB231303, Cambridge, UK), and vimentin (Cell Signaling, #9856S, Danvers, MA, USA) were added, and the cells were incubated overnight at 4 °C. The samples were then washed with DPBS, 10 min per wash. Next, secondary antibodies corresponding to the primary antibodies—goat anti-mouse IgG H&L (Jackson, #115-605-003, Lansing, MI, USA) and goat anti-rabbit IgG H&L (Jackson, #111-545-003, Lansing, MI, USA)—were applied. Finally, the nuclei were stained with Hoechst 33,342 solution (ThermoFisher, #62249, Waltham, MA, USA). The stained samples were imaged using a Confocal Laser Scanning Microscope (LEICA, TCS SP8, Wetzlar, Germany) to obtain fluorescence images. The fluorescence intensity of the images was analyzed using ImageJ software.

### 3.8. Statistics and Analysis

Statistical analyses were performed using Excel (Microsoft) and Prism 9.5 (GraphPad). The significance of differences in immunofluorescence intensity and spheroid invasion in the Matrigel embedding model was determined using a *t*-test. Differences between BPA concentrations and exposure times were assessed using one-way ANOVA. Differences were considered statistically significant when *p* < 0.05 (* *p* < 0.05, ** *p* < 0.01, and *** *p* < 0.001).

## 4. Conclusions

In this study, we successfully established 3D PTC spheroids and their ECM-embedded models and explored the possibility and causes of BPA-induced malignant progression of their cancers (Figure 13). Although TPC-1 and BCPAP are both PTC cell lines, their spheroid structures and the responses to BPA are still different. When exposed to the same concentration of BPA, the tolerance of the TPC-1 spheroid was higher than that of the BCPAP spheroid, with the latter appearing to have a loose appearance and a fragmented cytoskeleton. Therefore, the concentration of BPA that promotes the proliferation and invasiveness of cancer spheroid is within a range beyond which the cancer spheroid shows a significant decrease in cell viability or even apoptosis. In addition, the E-cadherin and vimentin expression of these two cell lines was also affected by the 2D or 3D culture environment, which again illustrates the importance of cell model selection. This study showed that BPA may increase dedifferentiation, EMT, and invasive ability in PTC cells. These findings are critical for developing clinical care and treatment strategies, especially for RAI (Figure 14). However, this study still has its limitations. Although the U.S. Environmental Protection Agency and the Food and Drug Administration have established a safe reference dose (RfD) of 50 μg/kg/day for human BPA [20], a correlation with the human exposure dose was not established in this study. In addition, the current research output and efficiency can be further enhanced. Through high-throughput analysis and the integration of image recognition technology, we expect to assist clinicians and researchers in drug therapy evaluation and hazardous substance toxicity assessment in the future, providing a more efficient platform for treating and preventing thyroid cancer.

## Figures and Tables

**Figure 1 ijms-26-00814-f001:**
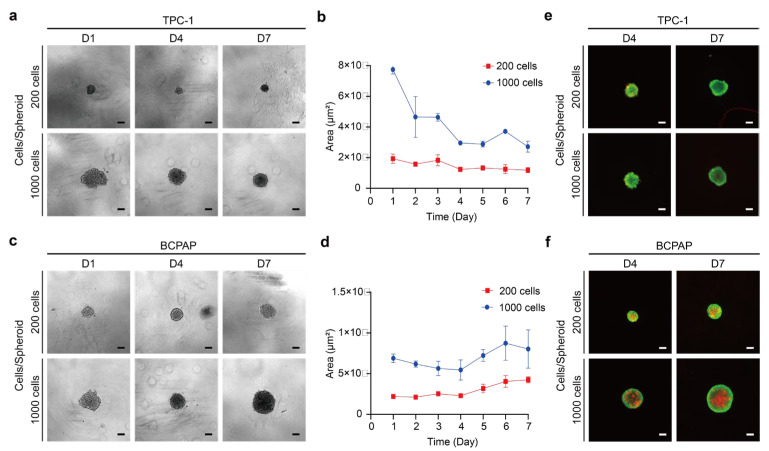
Spheroid formation and cell viability testing of TPC-1 and BCPAP spheroids. (**a**) Bright-field images of TPC-1 spheroids formed with cell densities of 200 and 1000 cells/well on days 1, 4, and 7. Scale bar: 100 µm. (**b**) Growth curve of TPC-1; the results were the means from three independent experiments, and the error bars indicate the standard errors. (**c**) Bright-field images of BCPAP spheroids formed with cell densities of 200 and 1000 cells/well on days 1, 4, and 7. Scale bar: 100 µm. (**d**) Growth curve of BCPAP; the results were the means from three independent experiments, and the error bars indicate the standard errors. (**e**) Cell viability was examined by Calcein-AM/PI staining. Fluorescence images show live (green) and dead (red) TPC-1 spheroid cells after 4 and 7 days of culture. Scale bar: 100 µm. (**f**) Fluorescence images show live (green) and dead (red) BCPAP spheroid cells after 4 and 7 days of culture. Scale bar: 100 µm.

**Figure 2 ijms-26-00814-f002:**
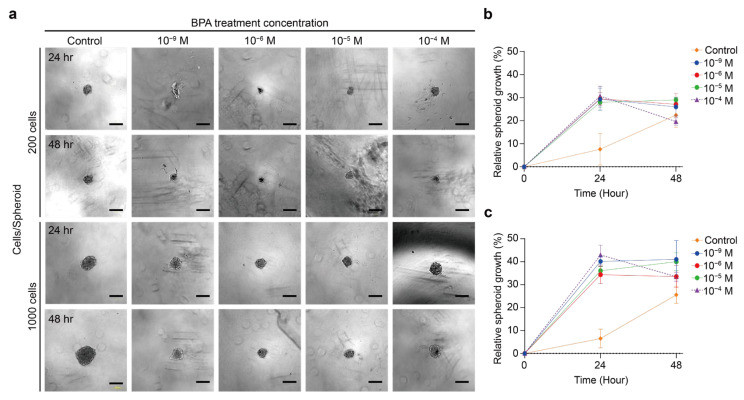
Cell proliferation of TPC-1 after treatment with BPA. (**a**) Bright-field images show TPC-1 spheroids treated with various concentrations (0~10^−4^ M) of BPA for 24 h and 48 h. Scale bar: 200 µm. (**b**) Line graph showing the relative growth area (%) of TPC-1 spheroids with a cell density of 200 cells/well after BPA treatment for 24 h and 48 h. The results were the means from three independent experiments, and the error bars indicate the standard errors. (**c**) Line graph showing the relative growth area (%) of TPC-1 spheroids with the cell density of 1000 cells/well after BPA treatment for 24 h and 48 h. The results were the means from three independent experiments, and the error bars indicate the standard errors.

**Figure 3 ijms-26-00814-f003:**
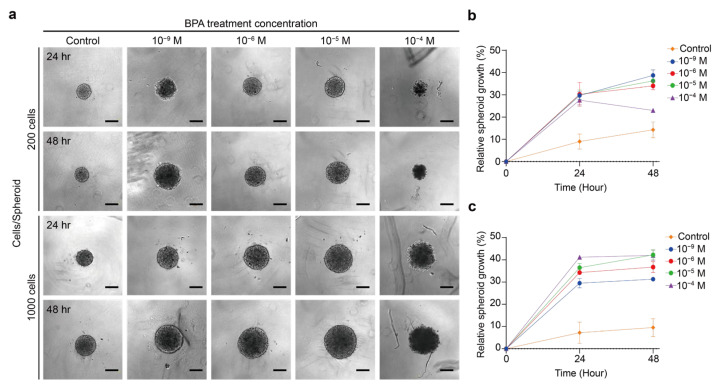
Cell proliferation of BCPAP after treatment with BPA. (**a**) Bright-field images show BCPAP spheroids treated with 0 to 10^−4^ M BPA for 24 h and 48 h. Scale bar: 200 µm. (**b**) Line graph showing the relative growth area (%) of BCPAP spheroids with a cell density of 200 cells/well after BPA treatment for 24 h and 48 h. The results were the means from three independent experiments, and the error bars indicate the standard errors. (**c**) Line graph showing the relative growth area (%) of BCPAP spheroids with t a cell density of 1000 cells/well after BPA treatment for 24 h and 48 h. The results were the means from three independent experiments, and the error bars indicate the standard errors.

**Figure 4 ijms-26-00814-f004:**
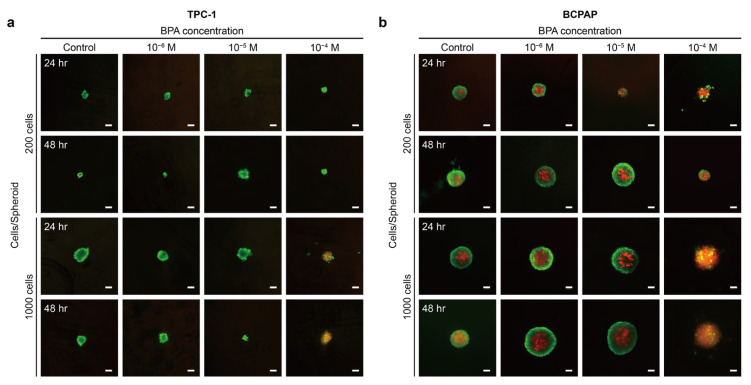
Cell viability of thyroid cancer cell line after treatment with BPA. (**a**) Fluorescence images showing live (green) and dead (red) TPC-1 spheroid cells after treatment with 0 to 10^−4^ M BPA for 24 h and 48 h. Scale bar: 100 µm. (**b**) Fluorescence images show live (green) and dead (red) BCPAP spheroid cells after treatment with 0 to 10^−4^ M BPA for 24 h and 48 h. Scale bar: 100 µm.

**Figure 5 ijms-26-00814-f005:**
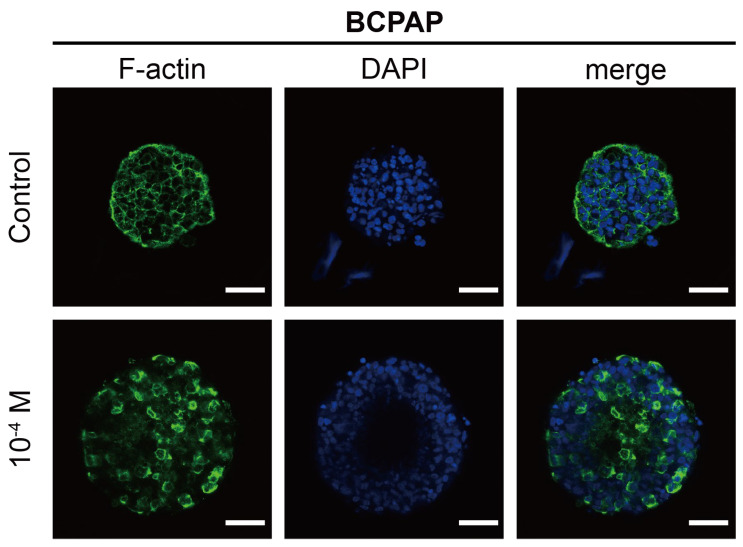
Immunofluorescence images showing the cytoskeletal arrangement in BCPAP spheroids after 48 h treatment with 10^−4^ M BPA. Spheroids were stained with Hoechst (blue) for nuclei and phalloidin (green) for F-actin. Scale bar: 50 µm.

**Figure 6 ijms-26-00814-f006:**
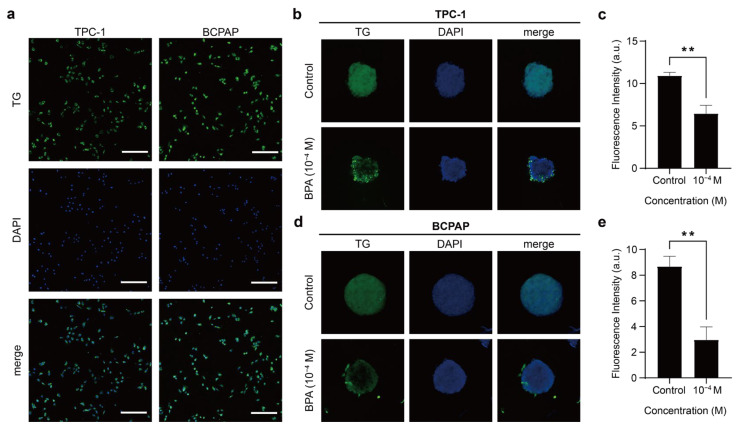
PTC expression of thyroglobulin (TG). (**a**) Immunofluorescence images showing TG expression (green) and DAPI-stained nuclei (blue) in TPC-1 and BCPAP cells cultured in monolayer. Scale bar: 250 µm. (**b**) Immunofluorescence images showing TG expression (green) and DAPI-stained nuclei (blue) in TPC-1 spheroids before and after 48 h treatment with 10^−4^ M BPA. Scale bar: 100 µm. (**c**) The fluorescence intensity of TG per unit area in TPC-1 spheroids, compared before and after treatment with 10^−4^ M BPA. The results presented in these histograms were the means from three independent experiments, and the error bars indicate the standard errors (** *p* < 0.01). (**d**) Immunofluorescence images showing TG expression (green) and DAPI-stained nuclei (blue) in BCPAP spheroids before and after 48 h treatment with 10^−4^ M BPA. Scale bar: 100 µm. (**e**) The fluorescence intensity of TG per unit area in BCPAP spheroids, compared before and after treatment with 10^−4^ M BPA. The results presented in these histograms are the means from three independent experiments, and the error bars indicate the standard errors (** *p* < 0.01).

**Figure 7 ijms-26-00814-f007:**
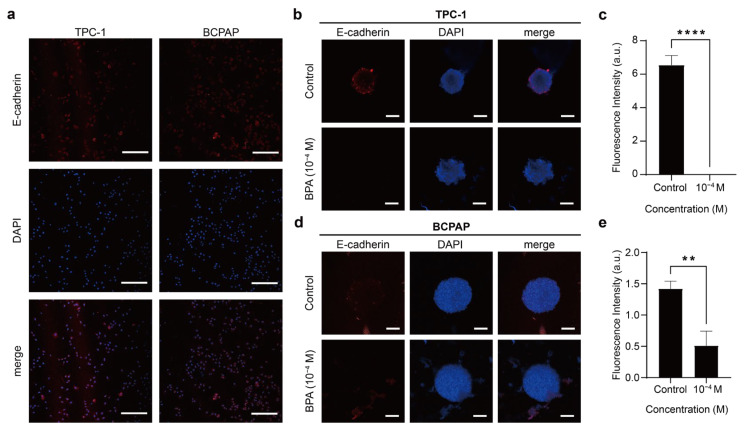
PTC expression of E-cadherin. (**a**) Immunofluorescence images showing E-cadherin expression (red) and DAPI-stained nuclei (blue) in TPC-1 and BCPAP cells cultured in monolayer. Scale bar: 250 µm. (**b**) Immunofluorescence images showing E-cadherin expression (red) and DAPI-stained nuclei (blue) in TPC-1 spheroids before and after 48 h treatment with 10^−4^ M BPA. Scale bar: 100 µm. (**c**) The fluorescence intensity of E-cadherin per unit area in TPC-1 spheroids, compared before and after treatment with 10^−4^ M BPA. The results presented in these histograms are the means from three independent experiments, and the error bars indicate the standard errors (**** *p* < 0.0001). (**d**) Immunofluorescence images showing E-cadherin expression (red) and DAPI-stained nuclei (blue) in BCPAP spheroids before and after 48 h treatment with 10^−4^ M BPA. Scale bar: 100 µm. (**e**) The fluorescence intensity of E-cadherin per unit area in BCPAP spheroids, compared before and after treatment with 10^−4^ M BPA. The results presented in these histograms were the means from three independent experiments, and the error bars indicate the standard errors (** *p* < 0.01).

**Figure 8 ijms-26-00814-f008:**
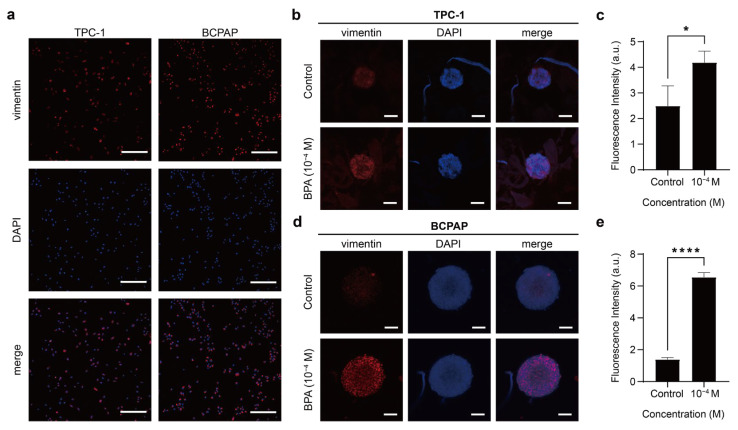
PTC expression of vimentin. (**a**) Immunofluorescence images showing vimentin expression (red) and DAPI-stained nuclei (blue) in TPC-1 and BCPAP cells cultured in monolayer. Scale bar: 250 µm. (**b**) Immunofluorescence images showing vimentin expression (red) and DAPI-stained nuclei (blue) in TPC-1 spheroids before and after 48 h treatment with 10^−4^ M BPA. Scale bar: 100 µm. (**c**) The fluorescence intensity of vimentin per unit area in TPC-1 spheroids, compared before and after treatment with 10^−4^ M BPA. The results presented in these histograms were the means from three independent experiments, and the error bars indicate the standard errors (* *p* < 0.05). (**d**) Immunofluorescence images showing vimentin expression (red) and DAPI-stained nuclei (blue) in BCPAP spheroids before and after 48 h treatment with 10^−4^ M BPA. Scale bar: 100 µm. (**e**) The fluorescence intensity of vimentin per unit area in BCPAP spheroids, compared before and after treatment with 10^−4^ M BPA. The results presented in these histograms were the means from three independent experiments, and the error bars indicate the standard errors (**** *p* < 0.0001).

**Figure 9 ijms-26-00814-f009:**
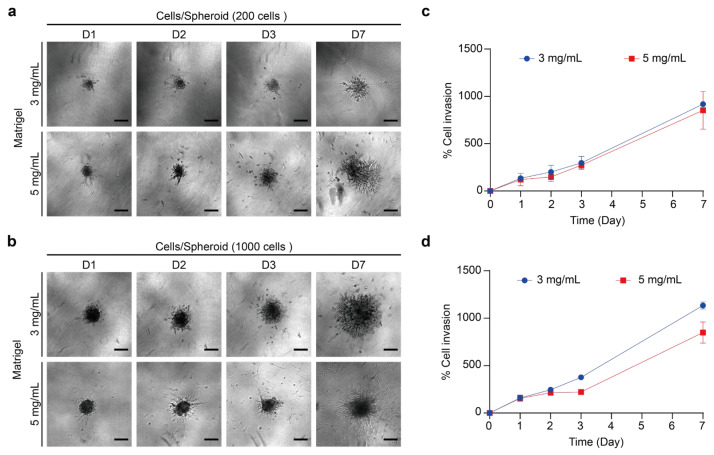
Concentration of Matrigel required for maintaining TPC-1 3D spheroid structure and invasion assay. (**a**) Bright-field images of spheroids with cell densities of 200 cells/well, embedded in 3 and 5 mg/mL Matrigel for 7 days. Scale bar: 200 µm. (**b**) Bright-field images of spheroids with cell densities of 1000 cells/well, embedded in 3 and 5 mg/mL Matrigel for 7 days. Scale bar: 200 µm. (**c**) Quantitative analysis of the invasion area (%) for cell densities of 200 cells/well relative to day 0. Results were the means from three independent experiments, and the error bars indicate the standard errors. (**d**) Quantitative analysis of the invasion area (%) for cell densities of 1000 cells/well relative to day 0. Results were the means from three independent experiments, and the error bars indicate the standard errors.

**Figure 10 ijms-26-00814-f010:**
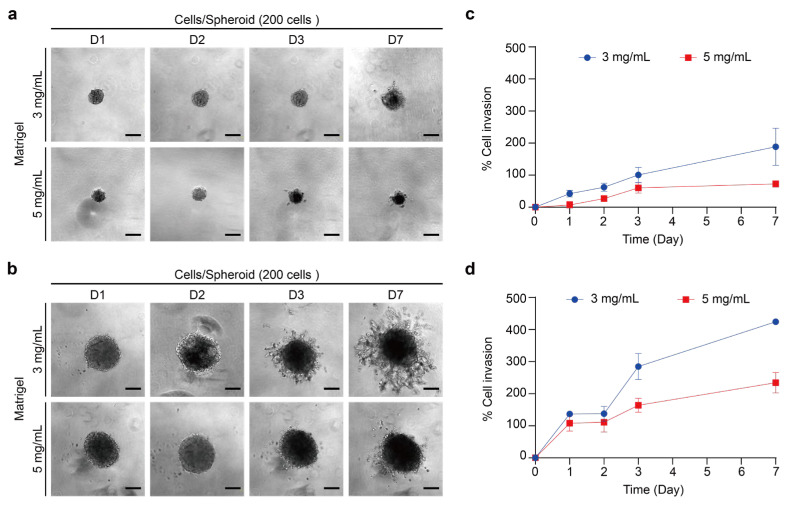
Concentration of Matrigel required for maintaining BCPAP 3D spheroid structure and invasion assay. (**a**) Bright-field images of spheroids with cell densities of 200 cells/well, embedded in 3 and 5 mg/mL Matrigel for 7 days. Scale bar: 200 µm. (**b**) Bright-field images of spheroids with cell densities of 1000 cells/well, embedded in 3 and 5 mg/mL Matrigel for 7 days. Scale bar: 200 µm. (**c**) Quantitative analysis of the invasion area (%) for cell densities of 200 cells/well relative to day 0. Results were the means from three independent experiments, and the error bars indicate the standard errors. (**d**) Quantitative analysis of the invasion area (%) for cell densities of 1000 cells/well relative to day 0. Results were the means from three independent experiments, and the error bars indicate the standard errors.

**Figure 11 ijms-26-00814-f011:**
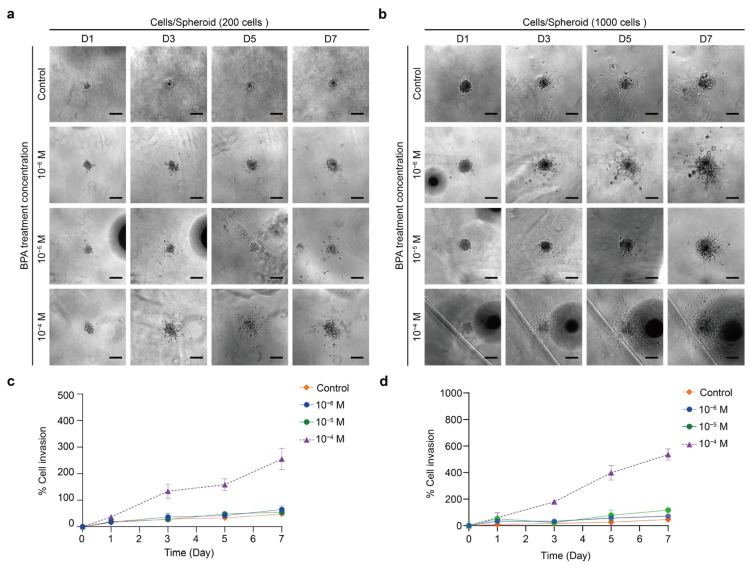
TPC-1 invasion assay across 0 to 10^−4^ M BPA treatment groups. (**a**) Bright-field images of spheroids with a cell density of 200 cells/well after 7 days. Scale bar: 200 µm. (**b**) Bright-field images of spheroids with cell densities of 1000 cells/well in 7 days. Scale bar: 200 µm. (**c**) Quantitative analysis of the invasion area (%) for cell densities of 200 cells/well relative to day 0. Results were the means from three independent experiments, and the error bars indicate the standard errors. (**d**) Quantitative analysis of the invasion area (%) for cell densities of 1000 cells/well relative to day 0. Results were the means from three independent experiments, and the error bars indicate the standard errors.

**Figure 12 ijms-26-00814-f012:**
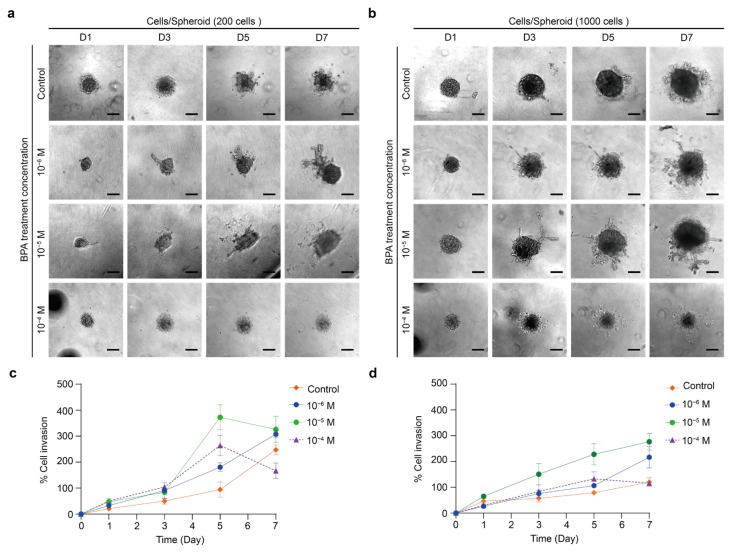
BCPAP invasion assay across 0 to 10^−4^ M BPA treatment groups. (**a**) Bright-field images of spheroids with a cell density of 200 cells/well after 7 days. Scale bar: 200 µm. (**b**) Bright-field images of spheroids with cell densities of 1000 cells/well in 7 days. Scale bar: 200 µm. (**c**) Quantitative analysis of the invasion area (%) for cell densities of 200 cells/well relative to day 0. Results were the means from three independent experiments, and the error bars indicate the standard errors. (**d**) Quantitative analysis of the invasion area (%) for cell densities of 1000 cells/well relative to day 0. Results were the means from three independent experiments, and the error bars indicate the standard errors.

**Figure 13 ijms-26-00814-f013:**
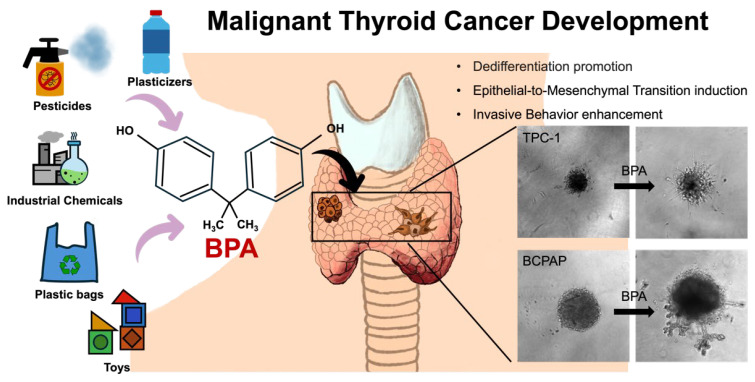
This study established a 3D in vitro thyroid cancer model to evaluate the effects of BPA exposure. The results revealed that BPA exposure promotes dedifferentiation of thyroid cancer spheroids, induces EMT, and enhances invasive behavior.

**Figure 14 ijms-26-00814-f014:**
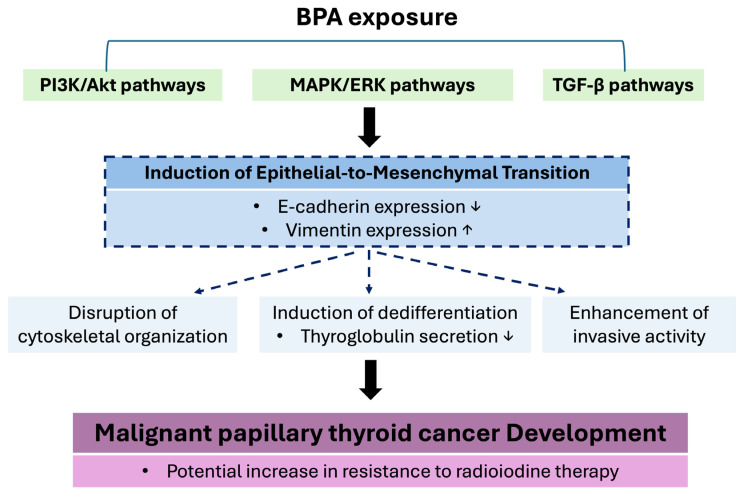
Conclusive image illustrating the impact of BPA on the malignant progression of thyroid cancer.

## Data Availability

Data will be made available on request.

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
