# Peer review of "Environmental Exposure to Bisphenol A Enhances Invasiveness in Papillary Thyroid Cancer"

_ijms, 2025, doi:10.3390/ijms26020814_

Round 1
Reviewer 1 Report
Comments and Suggestions for Authors
In the present study titled "Environmental Exposure to Bisphenol A Enhances Invasiveness in Papillary Thyroid Cancer," the authors investigated the role of BPA in the progression of papillary thyroid cancer. The study is well-designed and conducted; however, certain changes and additional experiments are necessary to strengthen the manuscript and make the findings more robust.
In Figures 2, 3, 11, and 12, control images are missing, although the corresponding graphs present the data. Authors should include these control images for better clarity and validation.
The term "in vitro" should be italicized throughout the manuscript to align with scientific conventions.
The authors should perform a western blot analysis for markers of invasiveness. This would help confirm the findings at the protein level and provide greater stability to the study.
What was the rationale for selecting MDA-T32, BCPAP, and TPC-1 cells? Additionally, did the authors include any non-cancerous cells to demonstrate the effect of BPA? Including such controls would strengthen the conclusions.
The quality of the figures is poor. The authors should replace all the figures with high-resolution versions to improve the overall presentation and readability of the manuscript.
Author Response
Comment 1: In Figures 2, 3, 11, and 12, control images are missing, although the corresponding graphs present the data. Authors should include these control images for better clarity and validation.
Response 1: Thank you for your kind reminder. We have respectfully added the control group photos to Figures 2, 3, 11, and 12 as suggested.
Comment 2: The term "in vitro" should be italicized throughout the manuscript to align with scientific conventions.
Response 2: Thank you for your kind reminder. We have thoroughly reviewed and revised the entire text. Please refer to the red text indicating the changes: Line 36, 128, 816.
Comment 3: The authors should perform a western blot analysis for markers of invasiveness. This would help confirm the findings at the protein level and provide greater stability to the study.
Response 3: Thank you for your valuable suggestion. We agree that performing Western blot analysis to detect invasion-related biomarkers (such as E-cadherin, vimentin, or MMPs) would help further confirm our findings at the protein level. The study by Li et al. utilized immunohistochemistry and Western blot analysis to examine tumor tissues from patients with papillary thyroid carcinoma (PTC), and the results showed a significant downregulation of E-cadherin expression, along with a marked increase in the protein levels of invasion markers such as N-cadherin and MMP-9 (J Cell Mol Med. 2021 Jan 19;25(3):1739–1749).
However, the current focus of this study is primarily on changes in cell behavior, and therefore, this analysis was not performed. Additionally, since the thyroid tumor spheroids in this experiment are not composed of a uniform cell layer, as shown in Fig. 4, where the intracellular and extracellular activities differ, and as seen in Fig. 7 and Fig. 8, which reveal differences in E-cadherin and vimentin expression between the inner and outer layers of cells, the core cells may exhibit different invasive characteristics due to hypoxia and nutrient deprivation, while the outer layer cells may be more exposed to the external environment and more sensitive to invasion-related signals. Western blotting requires cell lysis, a process that may lead to the averaging of results from cells with and without the target protein.
Therefore, we ultimately chose to use image analysis to obtain more representative data. We also recognize the importance of supplementing with molecular data and will consider employing other methods in future experiments further to strengthen the stability and reliability of our research conclusions.
Comment 4: What was the rationale for selecting MDA-T32, BCPAP, and TPC-1 cells? Additionally, did the authors include any non-cancerous cells to demonstrate the effect of BPA? Including such controls would strengthen the conclusions.
Response 4: This study selected the TPC-1, BCPAP, and MDA-T32 cell lines for experimentation, as these cell lines represent subtypes derived from different mutation sites in papillary thyroid carcinoma. We hypothesize that these mutations may be associated with the malignant progression induced by BPA exposure. Specifically, the TPC-1 cell line typically harbors the RET/PTC1 gene rearrangement, a mutation commonly observed in more invasive forms of papillary thyroid carcinoma (Virchows Archiv. 2013 July; 463: 437-444, Endocrine-related cancer. 2008 Mar; 15(1): 191). The BCPAP cell line carries the BRAF V600E mutation, which is frequently linked to the early onset of papillary thyroid carcinoma (J Clin Invest. 2005 Mar 10;115(4): 1068–1081). Meanwhile, the MDA-T32 cell line often possesses RAS gene mutations, which are associated with tumor proliferation and development (Front Endocrinol (Lausanne). 2012 Nov; 3:133, The Journal of Clinical Endocrinology & Metabolism, 2015 Feb; 100(2): 243-252).
Regarding the inclusion of non-cancerous cells as controls, we acknowledge the importance of incorporating them to assess the impact of BPA on normal cells. However, the primary aim of this study is to investigate the effects of BPA on papillary thyroid carcinoma (PTC) cells, with a focus on cancer progression and malignant transformation. Future research will consider the inclusion of non-cancerous thyroid cells to provide a more comprehensive exploration of this issue.
We appreciate your valuable feedback. The additional information has been incorporated into the manuscript to help readers better understand the study design and rationale, and relevant literature has been cited to support our conclusions. Please refer to the red text indicating the changes: Line 147-155.
Comment 5: The quality of the figures is poor. The authors should replace all the figures with high-resolution versions to improve the overall presentation and readability of the manuscript.
Response 5: We thank the reviewer for this comment. Following the reviewer's suggestion, we have improved the quality of all figures by increasing their resolution to 300 ppi (pixels per inch), which is the standard resolution for high-quality scientific publications. This enhancement has significantly improved the overall clarity and readability of the figures in the manuscript.
Reviewer 2 Report
Comments and Suggestions for Authors
Dear Authors,
below I present some minor corrections that would be beneficial to implement in your paper:
- Line 51, introduction – citation is needed
- Sentence starting from line 55 – this one should appear quite earlier in the text before providing more detailed information about the effects of BPA on the endocrine system
- Line 60 – reference is needed
- Figure 13 – it would be beneficial if you would improve the quality of the sentences (only them) because they are a little bit blurred
Big congratulations to your research group for providing such important results and conducting such novel research. More research in this area is still needed because the topic is not very well explored at this moment.
I wish you all the best with your further research
Kind regards
Author Response
Comment 1: Line 51, introduction – citation is needed
Response 1: Thank you for your valuable feedback. We have added a citation in Line 51 and marked it in red text.
Comment 2: Sentence starting from line 55 – this one should appear quite earlier in the text before providing more detailed information about the effects of BPA on the endocrine system
Response 2: Thank you for your valuable feedback. We have rearranged the order of the sentences.
Comment 3: Line 60 – reference is needed
Response 3: Thank you for your valuable feedback. We have added a citation in Line 60 and marked it in red text.
Comment 4: Figure 13 – it would be beneficial if you would improve the quality of the sentences (only them) because they are a little bit blurred
Response 4: Thank you for your valuable feedback. We have revised the caption for Figure 13 (Lines 816-818) to aid readers' understanding, and have marked the changes in red text.
Reviewer 3 Report
Comments and Suggestions for Authors
I have reviewed the manuscript “Environmental Exposure to Bisphenol A Enhances Invasiveness in Papillary Thyroid Cancer” submitted by Chien-Yu Huang et al., for publication in IJMS. The three-dimensional (3D) tumor spheroid model used in the study is an advanced approach to better simulate the tumor microenvironment compared to 2D cultures, providing more physiologically relevant results. I consider the experiments well-designed and address a critical topic: the impact of environmental contaminants on the development of cancers such as papillary thyroid carcinoma (PTC).
Major comments
Although the 3D spheroids are representative, data validating the findings in animal models or human clinical samples are not presented. I understand that animal models may be complex; however, analyzing tissue biopsies from BPA-exposed patients or serum samples could complement the authors' findings. Consider this.
The levels of BPA used should be contrasted with actual human exposures to ensure environmental relevance.
The differential response between the TPC-1 and BCPAP cell lines requires a broader discussion on how their genetic or epigenetic characteristics influence the results.
The discussion on BPA's potential to induce epithelial-mesenchymal transition (EMT) and increase cell invasiveness should relate more deeply to specific molecular mechanisms or signaling pathways. A conclusive image could help.
I consider it necessary to analyze the signaling pathways (PI3K/AKT, MAPK) activated by BPA, specifically concerning EMT.
Does BPA exposure affect the efficacy of standard therapies, such as radioiodine?
What would be the feasibility of conducting experiments with chronic exposure to low concentrations of BPA to simulate prolonged environmental exposures?
Analyze the effect of BPA on the interaction of tumor cells with tumor-associated fibroblasts or immune cells in co-cultures, in the context of a three-dimensional microenvironment resembling in vivo tumors.
What other EMT-related proteins, besides E-cadherin and vimentin, might be modulated by BPA and should be investigated? Discuss this.
Author Response
Comment 1: Although the 3D spheroids are representative, data validating the findings in animal models or human clinical samples are not presented. I understand that animal models may be complex; however, analyzing tissue biopsies from BPA-exposed patients or serum samples could complement the authors' findings. Consider this.
Response 1: Thank you for your valuable feedback. We appreciate your suggestion regarding the use of animal models or human clinical samples to validate our research findings. We have incorporated relevant references and content into the manuscript to strengthen the reliability of our conclusions, marked in red. In our future experimental design, we will consider integrating clinical samples to validate the results of our 3D spheroid model in a more physiologically relevant context.
Comment 2: The levels of BPA used should be contrasted with actual human exposures to ensure environmental relevance.
Response 2: Thank you for your valuable feedback. We have added the following information to the manuscript to enhance the persuasiveness of the experiment, and have marked the changes in red text (Line 226-240).
The study by Wang et al. reports that the corresponding exposure dose level of BPA in the general population is < 0.1 μg (kg bw)−1 day−1, which, based on an average adult weight of 70 kg, is approximately 10⁻⁸ M (Environmental Research. 2022 Oct; 213(1): 113734). In the research conducted by Li et al., the plasma BPA exposure level in the study subjects was 10⁻⁷ M, with no significant differences observed between the healthy control group and PTC patients, suggesting that this exposure level is representative of common human exposure (Journal of Cellular and Molecular Medicine. 2021 Jan; 25: 1739-1749). According to Table S2 compiled by Zhang et al., the BPA concentration range in the urine/blood of adult women and newborns is approximately 3.46 × 10⁻⁸ M to 5.65 × 10⁻⁹ M (Journal of Hazardous Materials. 2022 Mar; 425: 127911).
In addition, the BPA exposure encountered in daily life is typically at low doses with long-term exposure, such as gradual accumulation through plastic containers or food packaging. However, in experimental settings, we need to use BPA doses higher than those encountered in daily life to simulate and accelerate the cumulative effects of long-term exposure, which helps in the rapid evaluation of BPA's impact on cells or tissues. Therefore, based on the experimental parameters from previous literature (Journal of Cellular and Molecular Medicine. 2021 Jan; 25: 1739-1749, Journal of Hazardous Materials. 2022 Mar; 425: 127911, Endocrinology. 2009 Feb; 150(6): 2964-2973, Environmental Toxicology. 2024 Mar; 39(5): 3264-3273), this study employs BPA treatment concentrations ranging from 10⁻⁴ to 10⁻⁹ M, which encompass the BPA levels found in human urine or blood. On the other hand, the benchmark dose approach can be used to assess the potential effects of low-dose BPA exposure on human health (International Journal of Hygiene and Environmental Health. 2024 Jan; 255: 114293, Frontiers in Pharmacology. 2022 Feb; 12: 754408, International Journal of Hygiene and Environmental Health. 2024 Jan; 225: 114293, EFSA Journal. 2017 Jan; 15(1): 4658).
Comment 3: The differential response between the TPC-1 and BCPAP lines requires a broader discussion on how their genetic or epigenetic characteristics influence the results.
Response 3: We appreciate your suggestion and will include this discussion in the revised version, marked in red text (Line 664-692).
We agree that the differential responses between the TPC-1 and BCPAP cell lines are an important issue that warrants further discussion. In our manuscript, we will expand on this point and discuss the genetic and epigenetic factors that may contribute to the observed differences in response. This will help provide a more comprehensive understanding of how these cell lines respond to treatment and how their molecular characteristics may influence the experimental outcomes.
Regarding genetic mutations, the genes most commonly affected in papillary thyroid carcinoma (PTC) are BRAF and RET, which are activated through point mutations or chromosomal rearrangements (Endocrinology. 2007 Mar;148(3):936-41). The TPC-1 cell line typically harbors the RET/PTC1 gene rearrangement, a mutation frequently observed in more aggressive forms of PTC (Virchows Arch. 2013 Sep;463(3): 437-44, Endocr Relat Cancer. 2008 Mar;15(1): 191-205, Mol Cancer. 2010 Oct; 18: 9:278). The RET gene mutation leads to sustained activation of downstream signaling pathways, particularly the MEK/ERK and PI3K/AKT pathways, often resulting in abnormal cell proliferation, survival, and anti-apoptotic effects (Front Genet. 2021 Jan; 11: 618966). In contrast, the BCPAP cell line contains the BRAF V600E mutation. This mutation causes persistent activation of the MAPK/ERK pathway, which promotes cell proliferation and growth, and is commonly associated with the early onset of PTC (J Clin Invest. 2016 Apr;126(4): 1603). Moreover, this mutation induces the upregulation of tissue inhibitor of metalloproteinases-1 (TIMP-1), thereby facilitating PTC proliferation and metastasis (Virchows Arch. 2013 Sep;463(3): 437-44). However, unlike the RET mutation, the BRAF V600E mutation primarily affects the MEK/ERK pathway (Onco Targets Ther. 2018 Oct; 11: 7095-7107, Drug Des Devel Ther. 2014 Jun; 8:775-87), without directly impacting the PI3K/AKT pathway (Cell J. 2017 Winter;18(4): 485-492, Front Oncol. 2022 Jul; 12: 932353, Mol Cell Endocrinol. 2017 Dec; 457: 20-34).
Furthermore, epigenetic abnormalities are present in almost all cancers and, in conjunction with genetic alterations, promote tumorigenesis (Front Endocrinol (Lausanne). 2012 Mar; 3: 40). In thyroid cancer cells, epigenetic modifications encompass DNA methylation, histone modifications, and the expression of non-coding RNAs, with therapeutic agents targeting the first two modifications currently under clinical investigation (Sci Rep. 2018 Jul 27;8(1): 11315, Mol Cell Endocrinol. 2017 Dec; 457: 20-34). BRAF mutations, in particular, are closely linked to the abnormal methylation of multiple tumor suppressor genes, including tissue inhibitor of metalloproteinases-3 (TIMP3). Research suggests that the methylation-induced silencing of TIMP3 plays a critical role in the invasion and progression of PTC driven by BRAF mutations (Front Endocrinol (Lausanne). 2012 Mar; 3: 40). Additionally, PTEN, a phosphatase that inactivates the PI3K/Akt pathway, exhibits abnormal methylation in approximately 50% of papillary thyroid carcinomas and almost all follicular carcinomas and adenomas, implicating its potential involvement in thyroid tumor development (Thyroid. 2006 Jan;16(1): 17-23). Given the potential epigenetic differences between TPC-1 and BCPAP cell lines, these variations may influence the cellular response to BPA, resulting in distinct gene expression profiles that alter the effects of BPA.
Comment 4: The discussion on BPA's potential to induce epithelial-mesenchymal transition (EMT) and increase cell invasiveness should relate more deeply to specific molecular mechanisms nonsignaling pathways. A conclusive image could help.
Response 4: Thank you for your valuable suggestion. In the revised version, we plan to provide a detailed discussion on signaling pathways such as PI3K/Akt, MAPK, and TGF-β, and these will be highlighted in red (Line 693-704). Besides, Figure 14 (a conclusive image) is added. These pathways have been implicated in the effects of BPA on EMT, and their inclusion will enhance the discussion section of the manuscript.
Specifically, BPA influences the epithelial-to-mesenchymal transition (EMT) process through multiple signaling pathways, which interact and synergistically induce EMT, thereby increasing cell invasiveness and metastatic potential. First, BPA activates the PI3K/Akt signaling pathway, which plays a critical role in cell proliferation, survival, and migration. Overactivation of this pathway leads to an increase in matrix metalloproteinase (MMP) proteins, promoting matrix degradation and enhancing cell invasiveness (Toxicol Lett. 2014 Sep; 229(2): 357-65, Environ Sci Pollut Res Int. 2021 Apr; 28(16): 19643-19663, Arch Biochem Biophys. 2017 Nov; 633: 29-39). Additionally, BPA can activate the MAPK/ERK pathway, which induces changes in the cytoskeleton and the expression of MMPs such as MMP-2 and MMP-9, further strengthening cell migration and invasiveness (Toxicol Lett. 2014 Sep; 229(2): 357-65, Environ Sci Pollut Res Int. 2021 Apr; 28(16): 19643-19663, Reprod Toxicol. 2015 Dec; 58:229-33). Finally, BPA enhances the activity of TGF-β, a key EMT inducer, promoting the expression of EMT-related proteins like N-cadherin and vimentin, which facilitates the transition of cells from an epithelial to a mesenchymal phenotype (Exp Ther Med. 2021 Dec;23(2):164, Environ Pollut. 2021 Sep; 285:117472).
Comment 5: I consider it necessary to analyze the signaling pathways(PI3K/AKT, MAPK) activated by BPA, specifically concerning.
Response 5: Analyzing the PI3K/Akt and MAPK signaling pathways activated by BPA is crucial, as these pathways are key regulators of cell survival, proliferation, and migration. However, as the primary focus of this study is to examine the direct effects of BPA on specific cell models, we did not extend our analysis to these signaling pathways. Future research could benefit from exploring this aspect further. We appreciate your insightful suggestion and will incorporate this discussion into the revised version, with the addition highlighted in red (Line 705-713).
Research indicates that the MAPK and PI3K/Akt pathways are critical for cancer cell proliferation, survival, and migration (Cancer Lett. 2012 Jun; 319(1): 89-97, Hypertension. 2002 Nov; 40(5):748-54, Toxicol Lett. 2014 Sep; 229(2): 357-65). BPA can activate the PI3K/Akt pathway, enhancing these processes (Toxicol Lett. 2014 Sep; 229(2): 357-65, Environ Sci Pollut Res Int. 2021 Apr; 28(16): 19643-19663), and dysregulation of this pathway is common in thyroid cancer (Arch Biochem Biophys. 2017 Nov; 633:29-39, Environ Sci Pollut Res Int. 2021 Mar; 28(16): 19643–19663). BPA also modulates the MAPK pathway, promoting EMT and invasiveness (Environ Sci Pollut Res Int. 2021 Apr; 28(16): 19643-19663, Reprod Toxicol. 2015 Dec; 58: 229-33, J Cell Mol Med. 2021 Jan; 25(3): 1739–1749). For example, BPA activation of MAPK/ERK and PI3K/Akt pathways increases MMP-2, MMP-9, and N-cadherin expression, enhancing ovarian cancer cell migration (Toxicol Lett. 2014 Sep; 229(2): 357-65). Studying these pathways under BPA exposure could deepen understanding of cancer biology and identify potential therapeutic targets or biomarkers.
Comment 6: Does BPA exposure affect the efficacy of standard therapies, such as radioiodine?
Response 6: We appreciate your insightful suggestion and will incorporate this discussion into the revised version, with the addition highlighted in red (Line 429-442).
Currently, there is limited research regarding whether BPA exposure affects the efficacy of standard treatments, such as iodine-131 therapy. However, studies indicate that BPA may interfere with thyroid hormone signaling (J Clin Endocrinol Metab. 2002 Nov; 87(11): 5185-90, Reprod Toxicol. 2024 Oct; 129: 108680, Endocrinol Metab (Seoul). 2019 Dec; 34(4): 340–348), induce epithelial-mesenchymal transition (EMT) (Toxicol In Vitro. 2020 Feb; 62: 104676, J Cell Mol Med. 2021 Feb; 25(3): 1739-1749), and impact thyroid differentiation markers, such as the sodium/iodide symporter (NIS) (Front Endocrinol (Lausanne). 2024 Jul; 15: 1420540, Environ Int. 2019 May; 126: 321-328). Moreover, BPA may inhibit iodine uptake by interacting with NIS (Front Endocrinol (Lausanne). 2024 Jul; 15: 1420540). Since iodine-131 (I-131) therapy relies on NIS to transport radioactive iodine into thyroid cells for the destruction of hyperactive thyroid tissue or thyroid cancer cells (Pharmacol Ther. 2012 Sep; 135(3): 355-370, Nucl Med Mol Imaging. 2010 Feb; 44(1): 4–14).
Therefore, we hypothesize that if BPA alters NIS function, it could potentially reduce iodine-131 uptake efficiency, leading to therapeutic resistance in thyroid cancer cells and affecting treatment precision. Future studies are needed to explore the interaction between BPA and iodine-131 therapy to assess these potential effects.
Comment 7: What would be the feasibility of conducting experiments with chronic exposure to low concentrations of BPA to simulate prolonged environmental exposures?
Response 7: Thank you for your valuable suggestion. While our thyroid tumor spheroid model maintains stability for about 7 days, limiting long-term exposure studies, we recognize the importance of chronic low-dose BPA research. However, prolonged BPA exposure in 3D spheroids may pose challenges due to potential changes in proliferation and structure over time.
Currently, the development of thyroid cancer chips is limited, and their maintenance duration typically lasts only about one week. Daniel J et al. integrated flow conditions into a thyroid follicle chip device, successfully replicating the 3D structure of the natural microenvironment, the functionality of thyroid cancer, and the blood flow dynamics. After seven days of fluid perfusion, the study demonstrated that under dynamic conditions, the TG, NIS, DUOX genes, and cellular morphology were stably maintained. The average size of cell clusters significantly increased, and oxygen levels remained stable under dynamic conditions (Adv Healthc Mater. 2023 Mar; 12(8): e2201555).
Nonetheless, based on our previous experience with organ-on-chip platforms, these platforms have demonstrated excellent long-term culture stability, maintenance of physiologically relevant conditions, and the ability to monitor cellular responses over extended periods. For example, we developed an ALI (air-liquid interface) small airway chip in our previous research, which maintained functionality for up to 33 days (Biosensors (Basel). 2024 Nov; 14(12): 581).
Therefore, the organ-on-chip platform may offer valuable insights into the effects of chronic exposure on cells. We believe that transitioning this research to an organ-on-chip system could be a feasible approach for studying the long-term exposure to BPA in environmental contexts.
Comment 8: Analyze the effect of BPA on the interaction of tumor cells with tumor-associated fibroblasts or immune cells in co-cultures, in the context of a three-dimensional microenvironment resembling in vivo tumors.
Response 8: We appreciate your question and will include this discussion in the revised version, highlighted in red (Line 714-722).
Exposure to BPA within the tumor microenvironment may alter the interactions between tumor cells and stromal cells, such as fibroblasts and immune cells, thereby affecting tumor progression (Int J Mol Sci. 2024 Feb; 25(5): 2504), promoting an immunosuppressive environment (Cancers (Basel). 2022 Jun; 14(12): 3021, Int J Mol Sci. 2024 Jun; 25(11): 6259), and enhancing immune cell recruitment (Toxics. 2016 Sep; 4(4): 23, Sci Rep. 2020 Feb; 10(1): 3083, J Proteome Res. 2019 Dec; 19(2): 644–654). Specifically, BPA has been shown to facilitate the polarization of macrophages toward the M2 phenotype (Cancers (Basel). 2022 Jun; 14(12): 3021), reduce the recruitment of regulatory T cells (Toxics. 2016 Sep; 4(4): 23), and activate fibroblasts into tumor-associated fibroblasts (Cancers (Basel). 2022 Jun; 14(12): 3021, Biochim Biophys Acta Rev Cancer. 2024 Nov; 1879(6): 189190).
Analyzing the interactions between BPA, tumor cells, tumor-associated fibroblasts, or immune cells in co-culture, particularly within a 3D microenvironment that mimics in vivo tumor conditions, is crucial for understanding how BPA impacts the tumor microenvironment and tumor progression. We are considering incorporating additional stromal cells of the tumor microenvironment in future studies to explore these interactions further.
Comment 9: What other EMT-related proteins, besides E-cadherin and vimentin, might be modulated by BPA and should be investigated? Discuss this.
Response 9: We appreciate your inquiry and will include this discussion in the revised version, marked in red (Line 539-542).
In addition to E-cadherin and vimentin, proteins such as N-cadherin (J Cell Mol Med. 2021 Jan; 25(3): 1739–1749, Ecotoxicol Environ Saf. 2023 Oct; 264: 115479), fibronectin (Ecotoxicol Environ Saf. 2023 Oct; 264: 115479), Snail/Slug/Twist (Toxicol In Vitro. 2020 Feb; 62: 104676), MMPs (J Cell Mol Med. 2021 Jan; 25(3): 1739–1749, Toxicol Sci. 2015 Jul; 146(1): 101-15), and TGF-β (Drug Chem Toxicol. 2022 Sep; 45(5): 2285-2291, Chem Res Toxicol. 2015 Apr; 28(4): 662-71, Sci Rep. 2018 Jan; 8(1): 490) also play key roles in the EMT process of thyroid cancer. BPA may promote thyroid cancer progression and metastasis by modulating these proteins.
Round 2
Reviewer 3 Report
Comments and Suggestions for Authors
The authors have made significant changes to their manuscript and have adequately addressed my concerns. I believe the current version is suitable for publication.
Author Response
Dear Reviewer,
Thank you for your thoughtful and constructive feedback. We greatly appreciate your time and effort in reviewing our manuscript. Your comments were very helpful, and we are pleased to hear that the revised version meets your expectations.